# Enabling Older Adults’ Health Self-Management through Self-Report and Visualization—A Systematic Literature Review [note 1]

**DOI:** 10.3390/s20154348

**Published:** 2020-08-04

**Authors:** Gabriela Cajamarca, Valeria Herskovic, Pedro O. Rossel

**Affiliations:** 1Department of Computer Science, Pontificia Universidad Católica de Chile, Santiago 7820436, Chile; mgcajamarca@uc.cl; 2Department of Computer Science, Universidad Católica de la Santísima Concepción, Concepción 4090541, Chile; prossel@ucsc.cl; 3Centro de Investigación en Biodiversidad y Ambientes Sustentables (CIBAS), Universidad Católica de la Santísima Concepción, Concepción 4090541, Chile

**Keywords:** mHealth, monitoring, mobile, wearable, sensor, health, older, self-report, visualization

## Abstract

Aging is associated with a progressive decline in health, resulting in increased medical care and costs. Mobile technology may facilitate health self-management, thus increasing the quality of care and reducing costs. Although the development of technology offers opportunities in monitoring the health of older adults, it is not clear whether these technologies allow older adults to manage their health data themselves. This paper presents a review of the literature on mobile health technologies for older adults, focusing on whether these technologies enable the visualization of monitored data and the self-reporting of additional information by the older adults. The systematic search considered studies published between 2009 and 2019 in five online databases. We screened 609 articles and identified 95 that met our inclusion and exclusion criteria. Smartphones and tablets are the most frequently reported technology for older adults to enter additional data to the one that is monitored automatically. The recorded information is displayed on the monitoring device and screens of external devices such as computers. Future designs of mobile health technology should allow older users to enter additional information and visualize data; this could enable them to understand their own data as well as improve their experience with technology.

## 1. Introduction

Older adults are one of the fastest-growing demographic groups, generating a high demand for health, companionship and care services [1] that will only increase in the future. This increase in health service requirements in already overburdened healthcare systems could make it necessary for people to self-manage their health, for example through technology. *Mobile technology* refers to portable devices that users can carry with them (e.g., sensors, smartphones, or tablets) or wear. Mobile technology can be used to monitor, manage and motivate older adults in their care, for example, recording information such as the number of steps [2], heart rate [3], physical activity [4] or sleep quality [5] to assess health abnormalities and for the early detection of certain chronic diseases such as Parkinson’s disease or diabetes [6,7]. Mobile technology can help older adults maintain well-being and independence [8,9], for example, helping them change their eating behavior, physical activity, or emotional self-control [10,11,12], thus reducing the workload of health care providers, increasing efficiency, lowering health care costs, and improving patient comfort.

Even though older adults are often perceived as being wary of technology, some studies have found them to have a positive attitude towards it [13], finding benefits such as the possibility of supporting activities, adding convenience and being useful [13]. However, there is a digital divide, in which more privileged older adults (those with more education and higher income) have more Internet skills [14]. This, along with fear and anxiety when using technology, negative attitudes, feeling too old to learn, missing knowledge and experience, and not understanding digital terminology, can explain that some older adults feel resistance towards the use of technological tools [15]. Furthermore, the physical and neurodegenerative limitations [16] of older adults may make it necessary to create and design technologies to suit their specific needs, but in general terms, they are able to use almost any kind of technology, although sometimes with limitations. Regarding mobile technology, older adults are interested and have the intention to use it [17]. For example, most older adults are willing to use wearable devices, and the best predictor of their intention to do so is perceived usefulness [18].

On one hand, when mobile technology is used to monitor health, it is interesting to require the *input* of the user to complement data that may have been gathered automatically. The experience-sampling method asks users to provide self-reports of relevant data at random intervals [19]. For example, older adults can self-report pain [20] or monitor their blood glucose [7]. The input of information into a health monitoring device makes the user a more active participant in their own health, allowing them to complement automatically monitored data.

On the other hand, when data is monitored, a visualization of the data is usually *output*, or presented, to the users. Older adults have found visualizations to be useful to maintain awareness and recover from a bad day [21]. These visual representations of health data are important to maintain users’ self-awareness and interest in the health monitoring devices.

Previous studies on mobile technology for health monitoring of older adults have focused on mobile applications to assess balance as a risk factor for falls [22], on health monitoring in the home [23], on tools for the support of diseases or health conditions of older adults such as pain [17], cardiovascular disease [24], mental problems [25,26]. This systematic review aimed to collate studies of mobile technology designed for the health support of older adults, focusing on the input/output aspects of the technologies, that is, identifying whether these technologies allow older adults to visualize monitored information and whether they can enter complementary data such as context or feelings. We especially focus on older adults, since, as previously discussed, some of the consequences of aging relate to their capabilities when entering or visualizing information.

This paper is organized as follows. In Section 2, we describe the method used to locate and select studies for our review. In Section 3, we present the results as answers to our research questions. Then, we discuss our findings and present the conclusions.

## 2. Research Methods

This section describes the methodological framework used in this review, including the search strategy, search terms, selection of relevant articles, and data extraction.

### 2.1. Literature Search Strategy

A systematic search strategy was designed using the PRISMA guidelines [27], with the goal of having an overview of the mobile technology used for health monitoring of older adults. The research questions that guided the search and review were posed through the PICO format (population, intervention, comparison, outcome) [28]. The following three research questions were defined:

**RQ1** Which health information about older adults is usually monitored?**RQ2** What type of mobile technology is used by older adults to record and display health information?-Do mobile health technologies for older adults include self-reporting?-Do mobile health technologies for older adults include data visualization?**RQ3** How are mobile health technologies for older adults evaluated?

### 2.2. Search Terms

We defined our question of interest by using the three main PICO terms (Population, Intervention, and Outcome) (see Table 1). Our population of interest is *older adults*, who we consider to be anyone over 65 years old [29]. The intervention we are interested in studying is the use of mobile technology to record health information. The search includes features of the technology, such as the input of additional information (e.g., self-report) and data visualization, to evaluate the outcomes that they may have in health management.

The PICO terms allowed us to create a group of synonyms that were combined with the AND and OR operators to form a search string (Table 1). The asterisk operator (*) indicates that there may be more letters after the root word. We used this search string in four data sources: (1) Pubmed; (2) ACM Digital Library; (3) IEEE Xplore; and (4) Science Direct, by searching in publication titles, keywords and abstracts. The four data sources were selected to cover research related to health, computing, and technology.

### 2.3. Study Selection

The systematic search was carried out during November 2019. The search used the following eligibility criteria—(1) peer-reviewed articles, published in English between 2010 and 2019, (2) mobile technology (portable devices that users can carry with them) for monitoring the health of older adults, (3) mobile technology targeted at older adults to support the collection or display of health information. The exclusion criteria were that the technology was managed by third parties (caregivers, family, health professionals) and not by older adults, that the research did not present an evaluation, that the article was a research summary, invited plenary sessions, letters to the editor, or reviews, or if the article was not available for downloading.

### 2.4. Data Extraction

The analysis was carried out in a three-step process. First, the selected articles from all sources were organized in one file and duplicates were eliminated. Second, two reviewers (GC and PR) independently reviewed the titles and abstracts of the articles to decide whether an article met the inclusion criteria. Articles with two votes for inclusion or exclusion were automatically included or excluded. Disputed articles were resolved by a third reviewer (VH). Finally, two reviewers (GC and PR) read the full text and extracted relevant information.

## 3. Results

### 3.1. Reviewing Process

The first search of the four databases identified 609 publications. After eliminating duplicates, 417 results were left and of these, 240 were excluded based on a review of titles and abstracts. Subsequently, 177 articles were assessed for eligibility through full-text analysis and 103 publications met the inclusion criteria. Eight of these papers were shorter versions of other publications available in the corpus of identified publications, so the reviewers in a consensus discussion resolved not to include them in the results. In total, 95 publications were included in this report. The study selection process is depicted in Figure 1.

In the next subsections, we show the results in terms of 4 topics—(1) general study demographics including years and location, and the answers to each of our three research questions: (2) to answer RQ1, we describe which data is captured when monitoring the health of the older adults, (3) to answer RQ2, we describe characteristics of the monitoring technology (e.g., type, location, support), as well as the incorporation of self-reporting and visualization, and (4) to answer RQ3, we describe the methods used in the selected studies. These findings allow us to identify which mobile technologies are used to monitor and manage older adults’ health data and whether these technologies enable users to input additional information and visualize their health data.

### 3.2. General Study Characteristics

Figure 2 shows the number of publications by year and continent where the studies took place. Overall there is an increase in publications on mobile technology for monitoring the health of older adults in the 2009–2019 period. The number of studies increases from 3 in 2009 to 8 in 2014. In 2015, there is a decrease in publications (5), but the number of papers again increases from 2016 to 2019, reaching a maximum number of 19 studies in 2018. In terms of where the studies were conducted, 36 studies were done in North America (37.8%), 27 studies were conducted in Europe (28.4%), 13 in Asia (13.6%) and two in Latin America (2.1%). Most of the publications were presented as a journal article (55/95, 57.8%), followed by conference proceedings (24/95, 25.2%) and only 3 studies were part of a workshop (3.15%). The studies were published in the areas of health (57/95, 60%), computing (23/95, 24.2%) and 15 of them (15.8%) were presented in combined areas (i.e., health and computing).

### 3.3. Which Health Information about Older Adults Is Usually Monitored (RQ1)?

Table 2 summarizes the data that is monitored through mobile technology. The most common data that is captured is acceleration (58/95, 61.1%). Acceleration may be used to measure–or extrapolate–physical activity, activities of daily living (ADL), quality of sleep, or fall detection. The next type of frequently captured information is physiological parameters (26/95, 27.4%), including heart rate, blood pressure, foot pressure temperature, and blood oxygen saturation. Following this, we have process indicators (10/95, 10.5%) that are data related to the number of medications, interactions, messages, voltage, and memory skills. Other types of information such as sleep (8/95, 8,4%), emotions (5/95, 5.3%), position (4/95, 4.2%), pain (4/95, 4.2%), and time (4/95, 4.2%) were also recorded.

We identified several purposes for the collection of data—it was used (or expected to be used) for (1) general health monitoring (25/95, 26.3%), including monitoring feeding, sleep quality, physiological parameters, and medication management, (2) for gait detection (17/95, 17.9%), (3) fall monitoring (11/95, 11.6%), (4) physical activity (12/95, 12.6%), (5) activities of daily living (14/95, 14.7%), (6) rehabilitation (8/95, 8.4%), (7) mental health (6/95, 6.3%) and (8) social interaction (2/95, 2.1%).

### 3.4. What Type of Mobile Technology Is Used by Older Adults to Record and Display Health Information (RQ2)?

We identified 5 types of mobile technology used for older adults to monitor their health (Table 3)—(1) sensors, (2) smartphones, (3) tablets, (4) smart bracelets, and (5) tangibles. Sensors are the most frequently reported technology (60/95, 63.2%), and are used to record physiological information such as acceleration, heart rate, pressure, and strength. Sensors are devices that generally do not have their own interface and can function independently. Smartphones were used in 25.3% (24/95) of the reviewed studies. This technology allows the user to interact through touchscreen applications. Fourteen of the reviewed papers (14.7%, 14/95) presented a smart bracelet. These devices can include features such as an accelerometer, gyroscope, GPS, speaker, and screen. They also have connectivity mechanisms such as Bluetooth and WiFi. Some studies (12/95, 12.6%) presented applications to be used in a tablet device. Finally, technology with a tangible user interface was presented once (1/95, 1.1%). This technology refers to devices that allow users to interact with digital information through tangible objects.

Regarding where the technology is used, 12 locations were identified, including the hand (corresponding to handheld devices such as smartphones, tablets or mobile sensors) (38.9%), the wrist (24.2%), and the waist (17.8%). The least common location was the ear (2.1%).

#### 3.4.1. Do Mobile Health Technologies for Older Adults Include Self-Reporting?

Almost half (46/95, 48.4%) of the studies include methods for the older adults to enter health information. Table 4 provides an overview of how each of these 46 papers allows the input of health data. Twenty-nine of these studies (63%, (29/46)) used a device, and 22 (47.8%) used standard forms such as paper writing to enter data. Data reported by older adults to supplement data monitored by sensors or portable tools included physical, physiological, behavioral, mental, and subjective data. Smartphones and tablets were the most frequently used technology (14/29, 48.3%) while sensors and smart bracelets were the least frequently used technologies (9/29, 31.0%) to perform this action (see Table 5).

#### 3.4.2. Do Mobile Health Technologies for Older Adults Include Data Visualization?

Data collected from monitoring older adults’ health was displayed on screens in almost all studies (88/95, 92.6%). Out of these, 54 studies used the screen of the monitoring device to display information (61.3%), and 34 used external devices such as computers or tablets (38.6%). Of these 54 studies, 51 studies indicate that the display is intended for older adults (94.4%). However, only 12 of 51 studies (23.5%) evaluated characteristics of visualization such as clarity of information organization [30], straightforward understanding, and interpretation of displayed images [7], or the “look” of the visualizations (e.g., fonts, colors or size) [102]. In terms of data visualization using external devices, 4/34 articles (11.8%) focused on older adults, and among these 4 studies, only 2 studies (50%) evaluated aspects of visualization (see Table 6). One study evaluated compression of data displayed on the external device [113] and the other study compared visual feedback with animatronics (natural interaction) [72]. The technologies most frequently used by older adults to visualize health data were smartphones, sensors, and tablets. Smart bracelets were the least used (see Table 5).

### 3.5. How are Mobile Health Technologies for Older Adults Evaluated (RQ3)?

We studied how the mobile health technologies in the reviewed papers were evaluated. Out of the 95 studies, 58 applied quantitative methods (61.1%), 11 used qualitative methods (11.6%), and 26 used mixed methods, that is, both quantitative and qualitative methods (27.4%). The number of participants in the studies ranged from 3 to 3081 participants, with a mean of 101.3 and a median of 20. Only 9 studies had more than 100 participants. The average age of the participants in the studies was 63 years, with an age range from 23 to 86 years (median: 67). In general, most studies evaluated technology with older adult participants (63/95, 66.3%); only 5.3% (5/95) included young participants between 21 and 39 years old. Other studies (20/95, 21.1%) included both young and older adult participants. The rest of the studies (7/95, 7.4%) did not indicate the age of participants. Concerning mobile technology evaluation time, 49.5% of the reviewed publications reported that it ranged from 3 to 690 days, with a mean of 73 and a median of 42 days. Twelve studies reported evaluation periods shorter than 24 h. These shorter studies generally measured the accuracy of the technology. The remaining publications did not indicate the duration of the study.

Most of the reviewed studies recruited participants with underlying health conditions, for example, chronic conditions such as obesity and hypertension (8.4%) Parkinson’s disease (4.2%), chronic pain (3.2%), human immunodeficiency virus (HIV) (3.2%), and conditions causing cognitive impairment such as dementia, amnesic mild cognitive impairment or Alzheimer’s disease (6.3%). In only 15 studies (15.8%), the selected participants were generally healthy.

## 4. Discussion

Several works analyze the design and operational aspects of mobile health technology for older adults. For example, a recent literature review studied the possible barriers for older adults to use mobile technology [116], and another study evaluated the effects of technology-based self-monitoring that intend to improve health in middle-aged and older adults [117]. Other works on mobile technology have focused on specific aspects of older adult health, such as pain [17], dementia [26], or falls [22]. Unlike these studies, we present a systematic review of the literature on mobile health technology for older adults with a focus on self-reporting and data visualization. In other words, we identify whether the proposed technologies allow older adults to enter complementary health data and whether that information is presented to the users as an appropriate visualization.

The creation of devices that allow the entry of information is an opportunity to provide information from the patient’s point of view, information that sensors may not include. For example, self-report measures remain the most widely used methods for assessing sedentary behavior in adults [118]. Furthermore, older adults may become active participants in monitoring their health by entering data that is complementary to sensor data. Previous studies indicate that one way for older adults to gain autonomy and better engagement with sensor data is to use tools that allow for data annotation [119]. When designing devices that enable older users to enter health information, it should be considered that such information may be subject to error due to a variety of factors, including memory, cognitive ability, and awareness of the older user.

There is a lack of studies that consider how to deliver and display health information to older adults through mobile technology. Older adults must be included in the analysis of their own data so that they can contribute their contextual knowledge and so that the data can allow them to understand and self-reflect on their own behavior. Aside from the characteristics of delivering and displaying health information, research is needed on what information we should show to adult users. While some data such as heart rate, position, or acceleration are data used to detect anomalies in users’ health, some older users may be less experienced with the meaning of this information. In general, interpreting sensor data to obtain relevant information requires a process and knowledge [119]. In this regard, displaying appropriate information can be challenging, especially if different stakeholders have different information needs. Some authors indicate that older adults have different information needs concerning health care providers [120]. Besides, some studies (14/95, 14.7%) used more than one device to support health management of older adults (see Table 5). The use of several devices simultaneously can cause cognitive challenges [121] and lead to errors in the handling of technology by older adults [122]. In turn, these challenges lead to resistance by older people to use technology in daily life.

We found that the most frequently used location to use technology that monitors the health of older adults is the hand. The handheld devices include smartphones, tablets, and touchscreens. Regarding wearables, the most frequent placements in the body were the wrist, hip, or pocket. One study found that hand, wrist, forearm, upper arm, upper chest above the breast, forehead, ear, and mid thigh are the most appropriate locations for using wearable technology successfully [123]. Our review found that hand and wrist are often-used locations for wearables, while the ear was only used twice. Another study on wearable devices for older adults indicates that the wrist has higher acceptability than the arm and neck; however, personal attributes of gender, experience with technology, and need for medical care affect user preferences and attitudes [124].

Although technologies exist that allow for data entry and visualization of recorded information, there is not enough research on evaluating whether older adults are able to perform self-reporting and visualization actions in a way that is comfortable and easy for them. For example, data visualizations have been designed from a clinical perspective resulting in medical personnel being the only ones who can view and understand the monitored information. More research is needed on the design and evaluation of technologies that are appropriate for older adults, so that they are the ones who can self-manage and understand the recorded data.

Some limitations should be considered when assessing the results of this systematic review. We only included papers written in the English language; we excluded articles on mobile technology administered by health personnel, caregivers, or family members of older adults. We also chose to exclude studies that have not been published in peer-reviewed journals, conferences, or workshops, and we may have therefore missed out on some research and commercial solutions. By focusing on mobile technology that the older adult can carry with them, this review does not include health technologies that use static systems such as Kinect or fixed sensors and smart homes.

## 5. Conclusions

This systematic review found a growing number of research articles on mobile technology for health management in older adults in the 2009–2019 period. Most of the reviewed literature came from North America, Europe, and Asia (see Figure 2), and the format of presentations included journals, conferences, and workshop papers. Mobile technology used by older adults to record health information includes sensors, smartphones, and tablets. Typical monitored information includes acceleration, physiological parameters, position, sleep quality, pain levels, and emotions (see Table 2). The collection of these data has allowed researchers to monitor health and well-being issues such as activities of daily living, fall detection, gait detection, health care, mental health, physical activities, rehabilitation, and social interaction. The detection of such activities is essential to support aging in the home [23]. Particular attention should be paid to the development of mobile technology that allows adult users to input information. It is also essential to evaluate if the visualizations are adequate for older adults; in other words, it is necessary to include this population in the understanding of their own data.

Future work could focus on how to create appropriate visualizations; for example, to evaluate the effectiveness and understandability of each type of data that is presented to the users. It would also be interesting to study the types of visualizations that are provided (e.g., numbers, graphs), as well as whether the systems provide alerts when health data is not within normal ranges.

## Figures and Tables

**Figure 1 sensors-20-04348-f001:**
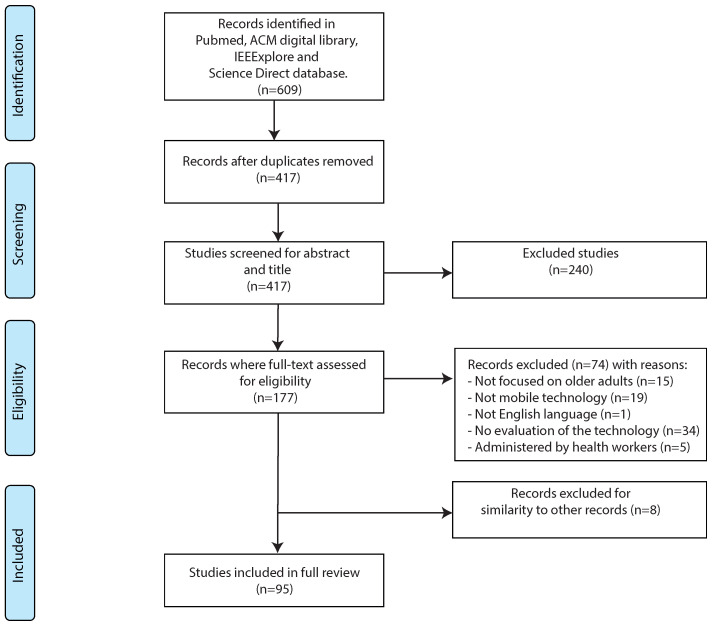
Flow-chart of the study selection process.

**Figure 2 sensors-20-04348-f002:**
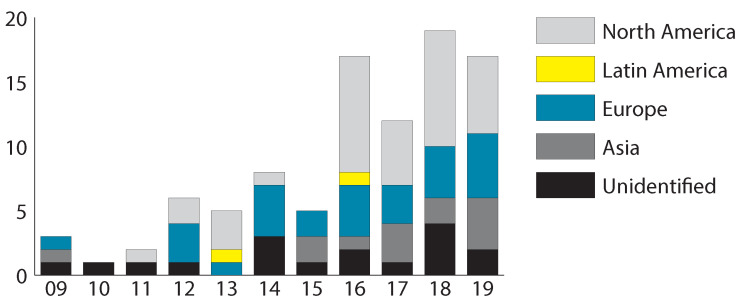
Frequency of publications per year.

**Table 1 sensors-20-04348-t001:** Search keywords aligned to PICO.

Variable	Description	Search Keywords
Population	Older adults	(older OR senior OR elder *) AND
Intervention	Mobile technology for health information	(mobile OR wearable OR sensor) AND health * AND
Outcome	Self-reporting and displaying	(feedback OR visualization OR self-report OR input)

**Table 2 sensors-20-04348-t002:** Data used for health monitoring of older adults.

	Studies	Count
Type of data		
Acceleration	[2,4,6,10,30,31,32,33,34,35,36,37,38,39,40,41,42,43,44,45,46,47,48,49,50,51,52,53,54,55,56,57,58,59,60,61,62,63,64,65,66,67,68,69,70,71,72,73,74,75,76,77,78,79,80,81,82,83]	58
Physiological parameters	[5,7,11,65,66,67,68,69,70,79,81,83,84,85,86,87,88,89,90,91,92,93,94,95,96,97]	26
Process indicators	[98,99,100,101,102,103,104,105,106,107]	10
Sleep	[5,71,79,82,83,94,108,109]	8
Emotions	[12,20,110,111,112]	5
Position	[75,76,77,78]	4
Pain	[20,111,112,113]	4
Time	[106,109,114,115]	4

**Table 3 sensors-20-04348-t003:** Type, Placement and features for mobile technology used to monitor the health of older adults.

	Studies	Count
Type of technology		
Sensor	[4,5,6,10,12,34,35,36,37,38,39,40,41,42,43,45,46,47,48,50,52,53,54,55,56,57,58,60,61,62,63,65,66,67,68,69,70,71,72,73,75,76,78,79,80,82,84,85,86,87,90,91,95,96,97,100,103,107,108,115]	60
Smartphone	[7,10,11,30,44,45,49,51,52,59,62,67,68,74,76,77,85,87,98,99,101,105,106,112]	24
Smart bracelet	[2,20,59,70,76,81,82,83,85,92,93,94,109,114]	14
Tablet	[31,32,33,64,88,89,102,104,108,110,111,115]	12
Tangible	[113]	1
Technology placement		
Hand	[7,10,11,30,31,32,33,44,45,49,51,52,53,64,68,74,76,77,85,87,88,89,90,91,98,99,101,102,104,105,106,108,110,111,112,113,115]	37
Wrist	[2,4,6,12,20,37,59,65,70,71,72,76,79,81,82,83,84,85,91,93,94,109,114]	23
Waist	[5,41,48,50,54,57,59,63,66,67,70,73,83,87,90,93,95]	17
Pocket	[39,40,42,51,54,57,59,68,79,88,91,92,110,112]	14
Foot	[6,45,63,66,69,80,95,96,97]	9
Chest	[46,48,56,60,63,65,67,85,86]	9
Back	[6,34,47,58,61,62,69,75,97]	9
Ankle	[6,38,53,55,65,73]	6
Neck	[35,43,44,74,100,103]	6
Legs	[48,58,64,66,75]	5
Finger	[84,87,107]	3
Ear	[78,84]	2

**Table 4 sensors-20-04348-t004:** Features of self-reporting for older adults

Self-report	Studies	Health Data
On device	[7,10,11,20,30,33,35,47,65,74,76,77,86,87,88,89,90,93,94,96,102,104,106,108,110,111,112,113,115]	Sleep quality, pain level, comfort, activity time, breathing rate, glucose level, stress level, activities attainment, pulses, weight, shortness of breath, feeling sad, vital signals, mood level, medication adherence, math tasks, falling number, fitness progress.
On paper	[5,6,35,36,57,60,62,64,70,71,74,81,82,83,93,94,98,99,103,105,109,114]	Physical activities, domestic tasks, physical activities, food quantity, activity energy, depression, memory, feeding speed, sleep quality, working time, fatigue level, perceived effort, talk time, domestic activities, free time, falling number, text message reply.

**Table 5 sensors-20-04348-t005:** The outcome of technology types, grouped by self-reporting and data visualization.

Type of Technology	Self-Report on Device	Visualization on Device
Smartphone	[7,11,30,74,80,106,112]	[7,11,30,44,49,51,74,77,98,99,105,106,112]
Sensors	[35,47,65,86,90,96]	[4,34,50,53,56,65,79,90,91,95,107]
Smart bracelet	[20,93,94]	[2,20,81,83,92,94,109,114]
Tablet	[33,88,89,102,104,110,111]	[31,32,33,64,88,89,102,104,110,111]
Tangible	[113]	NA
Sensor + Smartphone	[10,87]	[10,45,52,62,68,87]
Smart bracelet + Sensor	NA	[70]
Tablet + Sensor	[108,115]	[108,115]
Smart bracelet + Smartphone	NA	[59]
Smart bracelet + Sensor + Smartphone	[76]	[76,85]

**Table 6 sensors-20-04348-t006:** Visualization of older adults’ health data—specifications, studies, and count.

Visualization	Specifics	Studies	Count
On device		[2,4,7,10,11,20,30,31,32,33,34,44,45,49,50,51,52,53,56,59,62,64,65,68,70,74,76,77,79,81,83,85,87,88,89,90,91,92,94,95,98,99,102,104,105,106,107,108,109,110,111,112,114,115]	54
	Is it for older adults?	[2,4,7,10,11,20,30,31,32,33,34,45,49,50,51,52,53,56,59,64,65,68,70,74,76,77,79,81,83,87,88,89,90,91,92,94,95,98,99,102,104,105,106,107,108,109,110,111,112,114,115]	51
	Was it evaluated?	[4,7,11,30,31,33,51,52,68,89,102,106]	12
External		[5,6,12,35,36,38,39,40,41,42,43,47,48,55,57,58,61,63,67,69,72,73,75,78,80,82,84,86,93,96,97,100,103,113]	34
	Is it for older adults?	[35,58,72,113]	4
	Was it evaluated?	[72,113]	2

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
