# Peer review of "Enabling Older Adults’ Health Self-Management through Self-Report and Visualization—A Systematic Literature Review"

_sensors, 2020, doi:10.3390/s20154348_

Round 1

Reviewer 1 Report

This is an innovative and interesting manuscript with detailed and well-written introduction. A few questions that occurred to me are: is it necessary to include the statement "although in some countries with short life expectancy people over 50 are considered to be older adults"? This statement does not seem to really add anything to the paper. Second, is it important to provide the verbatim search string that was used in that much detail? The tables are excellent, detailed, and helpful. General study characteristics section is as well. Don't forget to spell out numbers and percentages when starting a sentence with these. I would encourage you to think carefully about perhaps more clearly identifying implications of your study and including more synthesis of your findings; as written, the discussion appears largely restating our results.

More specific comments are following:

p. 1 line 18-21, what do the authors mean by "The overload of these services could make it necessary. . . " This sentence seems contra to the following one where mobile technology is defined.
p. 3 line 80, why is it necessary to include the statement "in some countries with short life expectancy people over 50 are considered to be older"? This seems irrelevant.
p. 3 lines 85-86, is this much detail about the search string (and with the different font) needed?
p. 8 lines 239-, this section could be strengthened if there were more synthesis of the findings; what do they mean within the context of what is and isn't known about technology and older adults and health promotion? What are the implications of these findings in terms of practice? Research?

Reviewer 2 Report

Thank you for giving me the opportunnity tu review this interesting manuscript. I consider that the approach is correct, using the PRISMA guideline and the PICOS strategy in order to extract the information. However, I have some minor concerns in order to improve the quality of the manuscript.

  • Include the data where the search ended. 
  • Include in the flow chart the reasons why articles were excluded.
  • I would like to read which kind of outcomes/variables should incorpore these devices.
